# Age and Meniscal Extrusion Are Determining Factors of Osteoarthritis Progression after Conservative Treatments for Medial Meniscus Posterior Root Tear

**DOI:** 10.3390/jpm12122004

**Published:** 2022-12-03

**Authors:** Young Mo Kim, Yong Bum Joo, Byung Kuk An, Ju-Ho Song

**Affiliations:** 1Department of Orthopedic Surgery, Chungnam National University Hospital, Chungnam National University College of Medicine, Daejeon 35015, Republic of Korea; 2Department of Orthopedic Surgery, Chungnam National University Sejong Hospital, Chungnam National University College of Medicine, Sejong 30099, Republic of Korea

**Keywords:** meniscus root tear, non-operative treatment, arthritis

## Abstract

Background: With a growing understanding of biomechanical disadvantages following medial meniscus posterior root tear (MMPRT), recent studies have focused on surgical repair of MMPRT. Because not all tears are repairable, surgical indications can be properly established when the outcomes of conservative treatments are revealed. This study tried to identify risk factors for osteoarthritis progression after conservative treatments for isolated MMPRT. Materials & Methods: Patients who had conservative treatments for isolated MMPRT during 2013–2016 were retrospectively reviewed. To evaluate osteoarthritis progression, those who were followed up for ≤3 years and those who already showed advanced osteoarthritis of Kellgren--Lawrence (K-L) grade 4 at the time of diagnosis were excluded. Because patients with varus malalignment were candidates for realignment osteotomy, conservative treatments for MMPRT were applied to patients with well-aligned knees. Osteoarthritis progression was determined based on the K-L grading system, and risk factors including age, sex, body mass index, lower limb alignment, preoperative K-L grade, meniscal extrusion, and the presence of subchondral bone marrow lesion (BML) were analyzed using logistic regression analyses. Results: A total of 42 patients were followed up for 57.4 ± 26.8 months. During that period, osteoarthritis progression was noted in 17 (40.5%) patients. Based on univariate analyses for each risk factor, age, meniscal extrusion, and the presence of subchondral BML were included in the multivariate logistic regression analysis. The results showed that age (*p* = 0.028, odds ratio = 0.87) and meniscal extrusion (*p* = 0.013, odds ratio = 9.65) were significant risk factors. A receiver operating characteristic curve found that the cutoff age was 63.5 years, with the area under the curve being 0.72 (sensitivity, 68.0%; specificity, 70.6%). Conclusions: About two-fifths of patients who had conservative treatments for MMPRT underwent osteoarthritis progression in the mid to long term. Age and meniscal extrusion were determining factors of osteoarthritis progression. The risk for osteoarthritis progression was decreased when the age of patients was over 63.5 years.

## 1. Introduction

The posterior root attachment of the medial meniscus is a crucial component because root disruption causes a loss of hoop strain resistance of the meniscus [1,2]. Medial meniscus posterior root tear (MMPRT) is biomechanically equivalent to a total meniscectomy [3]. Unfortunately, MMPRT is not an uncommon disease, comprising up to 27.8% of all meniscal tears [4]. It is associated with degenerative changes in the knee joint, such as tibiofemoral chondral wear and meniscal extrusion [5,6,7].

Anatomic restoration of a ruptured root attachment using transtibial pull-out repair techniques is an ideal treatment for MMPRT [2,8,9]. In their matched cohort comparison study, Bernard et al. proved that meniscus root repair could lead to less osteoarthritis progression compared to nonoperative management and partial meniscectomy [9]. However, such repair is not always feasible due to substantial degeneration of meniscal tissue and concurrent osteoarthritis [10,11]. This leaves the question of what happens to the joint with MMPRT that is not included in the indication of surgical repair.

Osteoarthritis progression is the most important concern when meniscus root repair cannot be applied. If risk factors for osteoarthritis progression are identified, the indication of meniscus root repair should be expanded accordingly. Thus, this study tried to identify those risk factors in conservative treatments for MMPRT. It was hypothesized that certain preoperative factors were associated with osteoarthritis progression.

## 2. Materials & Methods

Patients who had conservative treatments for isolated MMPRT during 2013–2016 were retrospectively reviewed. MMPRT was defined as a complete radial tear within 9 mm of the root attachment. To evaluate osteoarthritis progression, those who were followed up for ≤3 years were not included in this study. Those who already showed advanced osteoarthritis of Kellgren--Lawrence (K-L) grade 4 at the time of diagnosis were also excluded, and surgical treatments such as arthroplasty and realignment osteotomy were considered for them. Because patients with varus malalignment were candidates for realignment osteotomy, conservative treatments for MMPRT were applied to patients with well-aligned knees (within −3° to 3° of the mechanical axis). Conservative treatments mainly consisted of nonsteroidal anti-inflammatory drugs and supervised muscle-strengthening exercises.

### 2.1. Study Design

In assessing osteoarthritis progression, standing radiographs at the initial visits and those at the latest visits were analyzed based on the K-L grading system: grade 1, doubtful narrowing of the joint space with possible osteophyte formation; grade 2, possible narrowing of the joint space with definite osteophyte formation; grade 3, definite narrowing of the joint space with moderate osteophyte formation; and grade 4, severe narrowing of the joint space with large osteophyte formation. Osteoarthritis progression was defined as the aggravation of K-L grade, which was not solely dependent on osteophyte formation [12].

To identify risk factors, age, sex, body mass index (BMI), lower limb alignment, preoperative K-L grade, meniscal extrusion, and the presence of subchondral bone marrow lesion (BML) were investigated. Regarding lower limb alignment, varus and valgus mechanical alignments were deemed positive and negative, respectively. Meniscal extrusion was assessed on preoperative magnetic resonance imaging (MRI) scans. A distance between two vertical lines touching the edges of the tibial plateau and the meniscus was measured, and meniscal extrusion exceeding 3 mm was counted (Figure 1A) [7,9]. Subchondral BML was defined as a locus of high signal intensity with trabecular marrow in it (Figure 1B) [13]. Radiographic evaluation was independently performed by two orthopedic surgeons, and all disagreements were resolved by discussion.

### 2.2. Statistical Analysis

Risk factors for osteoarthritis progression were identified using logistic regression analyses. To avoid overfitting problems, univariate analyses for each factor were performed before a multivariate regression analysis was conducted. Categorical variables were compared between the progression group and the non-progression group using the Chi-square test (Fisher’s exact test when the expected value of the cell was 5 or more in at least 80% of the cells), and continuous variables were compared using a *t*-test. A receiver operating characteristic (ROC) curve was used to determine the cutoff point of continuous variables that were significant risk factors based on the multivariate regression analysis. All statistical analyses were performed using R software version 4.1.1 (R foundation for Statistical Computing, Vienna, Austria), with a *p*-value < 0.05 considered statistically significant.

## 3. Results

A total of 42 patients were followed up for 57.4 ± 26.8 months. Their mean age was 63.4 ± 7.7 years. During the follow-up period, osteoarthritis progression was noted in 17 (40.5%) patients. The progression group and the non-progression group showed significant differences in age, meniscal extrusion, and the presence of subchondral BML. The patient characteristics according to osteoarthritis progression are presented in Table 1.

### Risk Factors for Osteoarthritis Progression

After univariate analyses for each risk factor, a multivariate logistic regression for osteoarthritis progression was performed with age, meniscal extrusion, and subchondral BML. The results showed that age (*p* = 0.028, odds ratio = 0.87) and meniscal extrusion (*p* = 0.013, odds ratio = 9.65) were significant risk factors (Table 2). A ROC curve found that the cutoff age was 63.5 years, with the area under the curve being 0.72 (sensitivity, 68.0%; specificity, 70.6%; Figure 2). 12 of 20 (60.0%) patients over the cutoff age underwent osteoarthritis progression whereas 5 of 22 (22.7%) patients under the cutoff age did.

## 4. Discussion

The most important finding of the present study was that age and meniscal extrusion were determining factors of osteoarthritis progression after conservative treatments for MMPRT. Age had a negative correlation with osteoarthritis progression, and the cutoff age was 63.5 years, meaning that patients over the cutoff age had a lesser risk of osteoarthritis progression. Meniscal extrusion exceeding 3 mm was also a risk factor. These factors should be considered when adopting conservative treatments for MMPRT.

MMPRT has been in the spotlight owing to the growing attention on meniscus root repair [5,14,15,16]. Such anatomic restoration has been endorsed by several biomechanical studies [3,17,18,19]. Allaire et al. demonstrated that MMPRT led to significant changes in contact pressure and knee joint kinematics, which were comparable to a total-meniscectomized state [3]. Marzo et al. also reported similar biomechanical results and argued that the surgical repair could restore the damaged biomechanics to within normal conditions [18]. The efficacy of meniscus root repair was supported by a clinical study comparing nonoperative management, partial meniscectomy, and repair. Bernard et al. performed a matched comparison among the different treatment options for MMPRT and proved that meniscus root repair led to significantly less osteoarthritis progression and subsequent knee arthroplasty [9]. It is evident that meniscus root repair should be attempted whenever possible. However, the problem is that the repair is not always feasible.

A large proportion of MMPRTs are degenerative lesions. As with the repair of other meniscal tears, meniscus root repair requires a robust remnant to place the sutures. Severe generation around the root attachment is a frequent reason why meniscus root repair cannot be achieved [15,20]. Lower limb alignment is another prerequisite for successful root repair [21,22]. If there is a varus malalignment, realignment osteotomy should be considered first. In a recent study investigating the treatment strategies of MMPRT, meniscus root repair was performed in less than 8.5% of the entire MMPRT patients [11]. The repair can be attempted only when a patient meets the narrow surgical indications [22]. Therefore, the prognosis of conservative treatments has clinical significance in establishing the approach for MMPRT.

This study found two risk factors for osteoarthritis progression: age and meniscal extrusion. It is noteworthy that age had a negative correlation with osteoarthritis progression. Although both biomechanical and clinical benefits are expected from meniscus root repair, it cannot be recommended readily to an older patient because complete healing is hard to expect after middle age [15,23]. Moreover, MMPRT is not an uncommon degenerative change, and other surgical options, such as arthroplasty and realignment osteotomy, are more frequently considered in older patients due to the severity of osteoarthritis [4,24]. According to the present study, the risk for osteoarthritis progression after conservative treatments is relatively low when the patient’s age is over 63.5 years. This result would help to establish the age-related indication of meniscus root repair.

Kwak et al. reported similar results on meniscal extrusion. They compared the patients who showed good responses to conservative treatment and the remaining patients who failed conservative treatment. It was concluded that the large meniscus extrusion ratio was the most reliable poor prognostic factor of conservative treatment for MMPRT [25]. Krych et al. performed an interesting study regarding the chronology of MMPRT. After reviewing serial MRI scans, they argued that meniscotibial ligament disruption and meniscal extrusion preceded MMPRT [14]. Their findings imply that meniscal extrusion is a separate degenerative change of the knee joint, which is in line with the conclusion of the present study. It can be assumed that meniscal extrusion is a risk factor for osteoarthritis progression because meniscal extrusion represents the degeneration of the meniscocapsular structure.

Several limitations should be noted. First, only a portion of MMPRT patients having conservative treatments was included, which might cause selection bias. Because of the well-documented disadvantages of MMPRT, the authors perform surgical repair or realignment osteotomy whenever possible. This study tried to evaluate the prognosis of patients who were not eligible for those surgical treatments. Second, the small sample size increased the risk of type 2 errors. This study only included patients who had been followed up long enough to evaluate osteoarthritis progression. Third, the majority of the included patients were female. Such gender specificity in MMPRT patients has been observed in the Asian population [20]. This should be considered when generalizing the conclusion of this study. Fourth, meniscal extrusion could not be assessed as a continuous variable because the sample size was not large enough.

## 5. Conclusions

About two-fifths of patients who had conservative treatments for MMPRT underwent osteoarthritis progression in the mid to long term. Age and meniscal extrusion were determining factors of osteoarthritis progression. The risk for osteoarthritis progression was decreased when the age of patients was over 63.5 years.

## Figures and Tables

**Figure 1 jpm-12-02004-f001:**
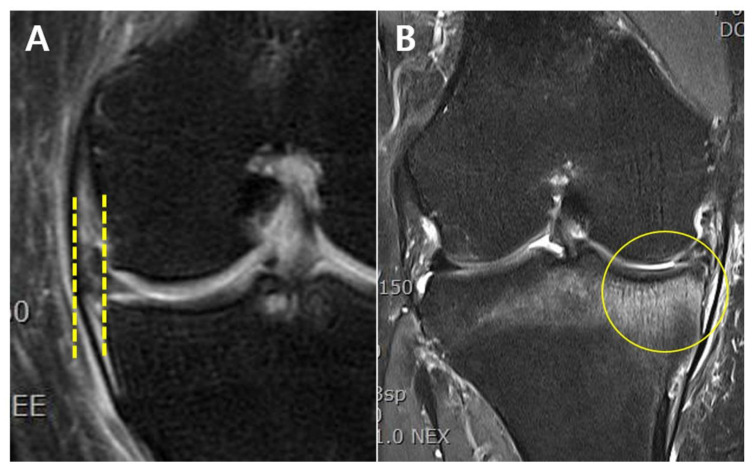
(**A**) Extrusion of the medial meniscus in a left knee. Meniscal extrusion was a distance between two vertical lines touching the edges of the tibial plateau and the meniscus. Meniscal extrusion exceeding 3 mm was counted. (**B**) Subchondral BML of the medial tibial plateau in a right knee. The lesion was a locus of high signal intensity with trabecular marrow in it. BML, bone marrow lesion.

**Figure 2 jpm-12-02004-f002:**
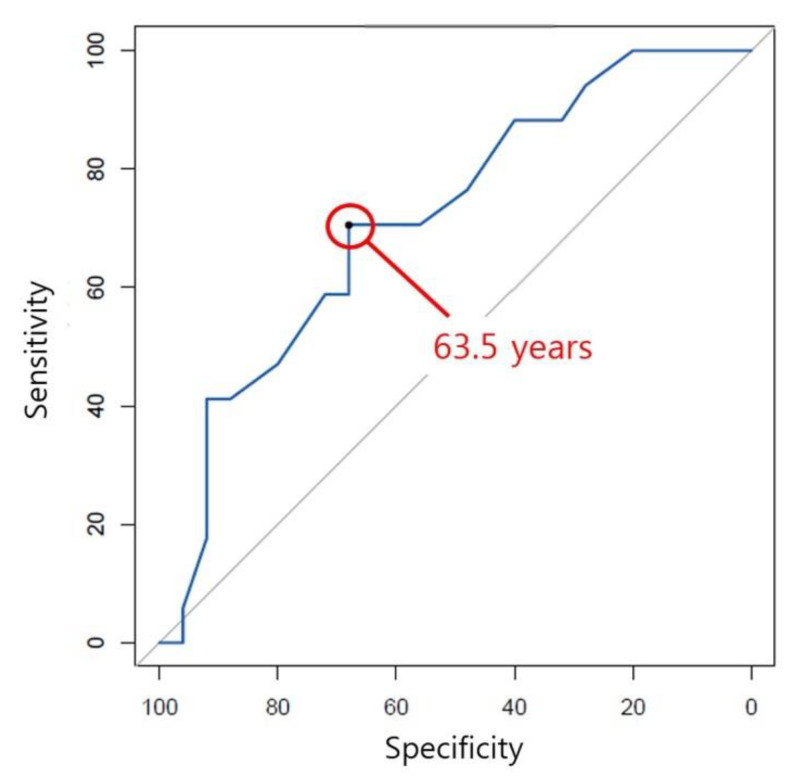
The receiver operating characteristic curve for osteoarthritis progression showed that the cutoff age was 63.5 years with the area under the curve being 0.72 (sensitivity, 68.0%; specificity, 70.6%).

**Table 1 jpm-12-02004-t001:** Patient characteristics according to osteoarthritis progression.

	Overall	Progression Group (N = 17)	Non-Progression Group (N = 25)	*p* Value
Age, yr	63.4 ± 7.7	59.9 ± 6.1	65.7 ± 7.9	0.015
Male/Female, n	38/4	17/0	21/4	0.134
BMI, kg/m^2^	25.5 ± 4.0	27.0 ± 5.2	24.4 ± 2.4	0.113
Follow-up duration, mo	57.4 ± 26.8	52.2 ± 26.0	60.3 ± 26.8	0.323
Lower limb alignment, deg ^a^	3.5 ± 2.2	3.2 ± 2.4	3.6 ± 2.1	0.698
Kellgren--Lawrence grade, n(grade 1/grade 2/grade 3)	32/9/1	12/4/1	20/5/0	0.558
Meniscal extrusion, n	14	11	3	0.001
Subchondral BML, n	12	8	4	0.041

BMI, body mass index; BML, bone marrow lesion. ^a^ Positive values indicate varus alignment, whereas negative values indicate valgus alignment.

**Table 2 jpm-12-02004-t002:** Logistic regression analyses regarding unstable flap.

	*p* Value		Exp(β) Coefficient (95% CI)	
	Univariate	Multivariate	Univariate	Multivariate
Age	0.023	0.028	0.90 (0.80–0.98)	0.87 (0.77–0.98)
Sex	0.999		0.99 (0.99–1.00)	
BMI	0.106		1.23 (0.96–1.58)	
Lower limb alignment	0.549		0.92 (0.69–1.22)	
Kellgren--Lawrence grade ^a^ (grade 1/grade 2/grade 3)	0.695		0.77 (0.21–2.81)	
Meniscal extrusion	0.001	0.013	13.44 (2.82–64.21)	9.65 (1.62–57.34)
Subchondral BML	0.035	0.118	4.67 (1.12–19.53)	4.53 (0.68–30.04)

BMI, body mass index; BML, bone marrow lesion. ^a^ Kellgren--Lawrence grade 2 and 3 were analyzed on the basis of grade 1.

## Data Availability

The data presented in this study are available on request from the corresponding author.

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
