# Peer review of "Age and Meniscal Extrusion Are Determining Factors of Osteoarthritis Progression after Conservative Treatments for Medial Meniscus Posterior Root Tear"

_jpm, 2022, doi:10.3390/jpm12122004_

Round 1
Reviewer 1 Report
In this retrospective study of the risk factors for OA
progression after conservative treatments for MMPRT ,
the authors reported that age and meniscal extrusion
are the determining factors. The manuscript is interesting and
and highlights the risk factors for MMPRT.The weak points
may be some of the methodology.
1) Lines 69-72 What was the definition of OA progression?
Did the authors try to determine OA progression by MRI
or computer assisted technique?
2) Line 66 What was the definition of well-aligned knees?
3) Lines 119,166,193 Will the cutoffs age change if the
patients undergo realignment osteotomy?
Author Response
In this retrospective study of the risk factors for OA progression after conservative treatments for MMPRT, the authors reported that age and meniscal extrusion are the determining factors. The manuscript is interesting and highlights the risk factors for MMPRT.The weak points may be some of the methodology.
1) Lines 69-72 What was the definition of OA progression?
The description has been added to the manuscript.
Did the authors try to determine OA progression by MRI or computer assisted technique?
We could not analyze using MRI or computer assisted technique which were not taken often in conservative treatments.
2) Line 66 What was the definition of well-aligned knees?
The description has been added to the manuscript
3) Lines 119,166,193 Will the cutoffs age change if the patients undergo realignment osteotomy?
It is well established that realignment osteotomy affects the prognosis of MMPRT. However, the comparison between well-aligned knees and surgically-corrected knees seems to be another interesting topic and is beyond the scope of this study.
Reviewer 2 Report
Dr. Kim's article is a valuable paper that aims to clarify one of the risks of knee OA progression after MMPRT.
This finding is clinically significant and interesting. I am satisfied with the quality of the work presented.
There have been several unclear issues, so I have a few questions and suggestions for corrections.
In lines 60-61, This is a retrospective study with a minimum of 3 years of follow-up. Is that an appropriate follow-up period? Compared to the proportion of patients with primary knee OA with disease progression over 3 years, do a high proportion of MMPRT patients progress? Please let me know if you have any information from previous studies.
In lines 80-82, A deviation of 3mm is judged as an Extrusion of the medial meniscus with reference to previous studies. Is it possible to analyze the extrusion distance as a continuous variable and perform a multiple regression analysis and ROC curve?
Author Response
Dr. Kim's article is a valuable paper that aims to clarify one of the risks of knee OA progression after MMPRT. This finding is clinically significant and interesting. I am satisfied with the quality of the work presented. There have been several unclear issues, so I have a few questions and suggestions for corrections.
In lines 60-61, This is a retrospective study with a minimum of 3 years of follow-up. Is that an appropriate follow-up period? Compared to the proportion of patients with primary knee OA with disease progression over 3 years, do a high proportion of MMPRT patients progress? Please let me know if you have any information from previous studies.
In this study, a minimum follow-up of 3 years was an inclusion criterion and the mean follow-up duration was 57.4 months, which was not so insufficient compared to previous studies.[1–3] Including the present study, there has been no study comparing the outcomes of conservative treatments in MMPRT and non-MMPRT. Because most previous studies investigated the prognosis of surgical procedures for MMPRT, the present study focused on OA progression after conservative treatments.
References
- Lim, H.C.; Bae, J.H.; Wang, J.H.; Seok, C.W.; Kim, M.K. Non-Operative Treatment of Degenerative Posterior Root Tear of the Medial Meniscus. Knee Surg Sports Traumatol Arthrosc 2010, 18, 535–539, doi:10.1007/s00167-009-0891-0.
- Lee, N.-H.; Seo, H.-Y.; Sung, M.-J.; Na, B.-R.; Song, E.-K.; Seon, J.-K. Does Meniscectomy Have Any Advantage over Conservative Treatment in Middle-Aged Patients with Degenerative Medial Meniscus Posterior Root Tear? BMC Musculoskelet Disord 2021, 22, 742, doi:10.1186/s12891-021-04632-8.
- Bernard, C.D.; Kennedy, N.I.; Tagliero, A.J.; Camp, C.L.; Saris, D.B.F.; Levy, B.A.; Stuart, M.J.; Krych, A.J. Medial Meniscus Posterior Root Tear Treatment: A Matched Cohort Comparison of Nonoperative Management, Partial Meniscectomy, and Repair. Am J Sports Med 2020, 48, 128–132, doi:10.1177/0363546519888212.
In lines 80-82, A deviation of 3mm is judged as an Extrusion of the medial meniscus with reference to previous studies. Is it possible to analyze the extrusion distance as a continuous variable and perform a multiple regression analysis and ROC curve?
The authors believe that the sample size is not large enough to analyze the extrusion distance as a continuous variable. This has been added in the limitation.
Round 2
Reviewer 1 Report
The authors have revised the manuscript well according to the reviewer’s comments .